# High-performance ZrNiSn-based half-Heusler thermoelectrics with hierarchical architectures enabled by reactive sintering

Xin Ai[1,2,9], Yu Wu [3,9], Haiyan Lyu[4,9], Lars Giebeler [1], Wenhua Xue[5], Andrei Sotnikov[1], Yumei Wang [5] ✉, Qihao Zhang[6], Denys Makarov [6], Yuan Yu [4], G. Jeffrey Snyder [7], Kornelius Nielsch [1,2,8] ✉ & Ran He [1] ✉

Half-Heusler compounds are promising thermoelectric materials for high-temperature applications, yet their performance is limited by high lattice thermal conductivity. Here, we present an alternative approach to synthesize ZrNiSn-based half-Heusler compounds with hierarchical architectures across multiple length scales. By utilizing short-duration mechanical alloying to produce nonequilibrium precursors, followed by reactive sintering, we enable precise control over phase composition and microstructural features. This approach results in multi-scale architectures comprising interstitial defects, grain boundaries, nanoprecipitates, and pores, enabling strong phonon scattering. The optimized $Zr_{0.75}Hf_{0.25}NiSn_{0.99}Sb_{0.01}$ alloy exhibits a lattice thermal conductivity as low as 1.9 W m$^{-1}$ K$^{-1}$ and a high power factor of 50 μW cm$^{-1}$ K$^{-2}$, yielding an impressive dimensionless figure of merit ($zT$) of 1.33 at 873 K. This performance surpasses that of ZrNiSn-based compounds synthesized via conventional methods such as arc melting and solid-state reaction. Our method, distinguished from conventional melting synthesis approaches through its simplicity, cost-effectiveness, and scalability, provides a versatile framework for achieving efficient hierarchical phonon scattering while preserving high carrier mobility in half-Heusler compounds and highlights the potential of reactive sintering for advancing thermoelectric materials.

Thermoelectric materials, which enable the direct conversion of heat into electricity, hold great promise for energy harvesting and waste heat recovery[1–3]. The efficiency of thermoelectric materials is quantified by the dimensionless figure of merit ($zT$), defined as $zT = \alpha^2\sigma T/\kappa$, where $\alpha$ is the Seebeck coefficient, $\sigma$ is the electrical conductivity, $T$ is the absolute temperature, and $\kappa$ is the total thermal conductivity.

The $\kappa$ encompasses contributions from electronic ($\kappa_e$), lattice ($\kappa_L$), and bipolar ($\kappa_{bip}$) thermal conductivities[4–6]. Achieving high $zT$ requires an optimization of these interdependent parameters to balance electrical and thermal transport properties[7–9].

Among various thermoelectric materials, half-Heusler compounds have attracted significant attention due to their excellent

[1]Leibniz Institute for Solid State and Materials Research (IFW) Dresden e.V., Dresden, Germany. [2]Institute of Materials Science, TUD Dresden University of Technology, Dresden, Germany. [3]Advanced Thermal Management Technology and Functional Materials Laboratory, Ministry of Education Key Laboratory of NSLSCS, School of Energy and Mechanical Engineering, Nanjing Normal University, Nanjing, China. [4]Institute of Physics (IA), RWTH Aachen University, Aachen, Germany. [5]Institute of Physics, Chinese Academy of Sciences, Beijing, China. [6]Helmholtz-Zentrum Dresden-Rossendorf e.V., Institute of Ion Beam Physics and Materials Research, Dresden, Germany. [7]Department of Materials Science and Engineering, Northwestern University, Evanston, IL, USA. [8]Institute of Applied Physics, TUD Dresden University of Technology, Dresden, Germany. [9]These authors contributed equally: Xin Ai, Yu Wu, Haiyan Lyu. ✉e-mail: wangym@iphy.ac.cn; k.nielsch@ifw-dresden.de; r.he@ifw-dresden.de

thermoelectric performance at high temperatures (≥600 °C), thermal stability, mechanical robustness, and low toxicity[10,11]. These materials exhibit high electrical conductivity in the range of $10^4$–$10^6$ S/m and power factors ($\alpha^2\sigma$) typically between 30–100 $\mu$W cm$^{-1}$ K$^{-2}$ over a broad temperature range, making them particularly suitable for high-temperature power generation[12,13]. However, the intrinsically high $\kappa_L$ of half-Heusler compounds limits their $zT$. For instance, undoped ZrNiSn compounds typically exhibit $\kappa_L$ values ranging from 5 to 10 W m$^{-1}$ K$^{-1}$, attributed to their high crystal symmetry and strong atomic bonding. Both restrict phonon scattering and stiffen lattice vibrations[14,15].

To overcome this limitation, extensive research has focused on enhancing phonon scattering to reduce $\kappa_L$ while preserving high power factors. Common approaches include nanostructuring strategies such as grain refinement, nanoprecipitates, and the creation of rich interfaces in nanocomposites[16,17]. While effective at reducing $\kappa_L$, these strategies often compromise electrical properties due to excessive scattering of charge carriers[3,18]. Defect engineering offers an alternative by manipulating point defects such as vacancy, substitution impurity, and interstitial atom[19–23]. These defects scatter high-frequency phonons, thereby reducing $\kappa_L$ through strain fields generated by lattice distortions. Despite these advances, the $\kappa_L$ of state-of-the-art half-Heusler compounds, such as ZrNiSn-based compounds, remains significantly higher than their theoretical minimum (~1.0 W m$^{-1}$ K$^{-1}$)[10,24].

In semiconductor thermal transport, phonons with a mean free path (MFP) spanning from several nanometers to a few hundred nanometers contribute cumulatively to $\kappa_L$, while charge carriers typically exhibit MFPs within just a few nanometers. This disparity underscores the importance of implementing hierarchical phonon scattering across all relevant length scales to reduce phonon lifetimes without degrading charge transport[25]. Significant improvements in $zT$ have been achieved by leveraging such hierarchical phonon scattering across multiple length scales, including atomic-scale lattice disorder, nanoscale endotaxial precipitation, and meso-scale grain boundaries, in PbTe- and GeTe-based compounds[25–28]. However, realizing efficient hierarchical phonon scattering in half-Heusler compounds is still challenging, possibly due to the high tolerance of half-Heusler compounds to defects caused by compositional variations[10,29,30]. The complexity lies not only in the need to precisely control and balance phonon scattering at atomic, nanoscale, and meso-scale levels but also in the lack of preparation methods for effectively realizing such architectures.

In this study, we present an alternative approach for synthesizing half-Heusler compounds with hierarchical architectures spanning multiple length scales. In the ZrNiSn-based half-Heusler system, we employ short-duration mechanical alloying (i.e., 6 h) of raw metal powders to produce non-equilibrium precursors, followed by reactive sintering, during which phase formation and densification occur simultaneously (Fig. 1a). This process strategically leverages energy pathways during diffusion and reaction to achieve phase purity and tailored defect configurations. By optimizing sintering parameters such as temperature and holding time, we achieve precise control over critical microstructural features, including interstitial defects, grain boundaries, nanoprecipitates, and pores. Consequently, we achieve a minimum lattice thermal conductivity of 1.9 W m$^{-1}$ K$^{-1}$ while maintaining a maximum power factor of 50 $\mu$W cm$^{-1}$ K$^{-2}$ (Fig. 1b), resulting in a remarkable $zT$ of approximately 1.33 at 873 K for Zr$_{0.75}$Hf$_{0.25}$NiSn$_{0.99}$Sb$_{0.01}$. This performance surpasses that of (Ti, Zr, Hf)NiSn-based compounds prepared using high-temperature self-propagation synthesis (SHS) and conventional methods, such as solid-state reaction, arc melting, and levitation melting (Fig. 1c)[16,17,22,30–35]. In addition to achieving superior thermoelectric properties, our work stands out by offering a simpler, more cost-effective, and highly scalable approach compared to conventional synthesis methods (Fig. S1). These advantages underscore the

transformative potential of reactive sintering for enabling efficient hierarchical phonon scattering and advancing the performance of half-Heusler thermoelectrics.

## Results

The synthesis of ZrNiSn-based half-Heusler materials is achieved through reactive sintering of mechanically alloyed precursor powders (Fig. S1b). The mechanically alloyed powders exhibit irregularly shaped particles ranging from submicron to ~4 $\mu$m in size, with the majority higher fraction of smaller particles (Fig. S2a), which can facilitate efficient diffusion during subsequent reactive sintering[36]. The X-ray diffraction (XRD) patterns of the precursors (Fig. S2b) indicate the presence of trace amounts of the half-Heusler phase and amorphous matter, demonstrating their non-equilibrium state. Reactive sintering was initially performed over a temperature range of 750 °C to 1100 °C, with 50 °C intervals and a holding time of 12 min, to obtain fully dense bulk samples (Table S1). Then, at the sintering temperature of 950 °C, the holding time was systematically reduced to 6, 4, and 2 min to produce porous bulk samples with relative densities of 98.2% (referred to as D98), 95.4% (D95), and 90.9% (D91), respectively. The XRD patterns of all bulk samples confirm the formation of a single-phase half-Heusler structure within the detection limit (Fig. S3)[37]. Backscattered electron (BSD) images and elemental mapping reveal that, with increasing sintering temperature, the ZrNi intermediate transforms into the half-Heusler phase due to a higher thermodynamic driving force (Fig. S4). At temperatures above 900 °C, a spotted pattern emerges due to the formation of Zr-rich half-Heusler phase domains, with incomplete homogenization driven by kinetic constraints. Grain size analysis reveals a temperature-dependent increase, with average values of 0.4, 0.6, and 0.9 $\mu$m for samples sintered at 850 °C, 950 °C, and 1050 °C, respectively (Fig. S5). A fractured surface morphology reveals a transition from intergranular fractures along grain boundaries at lower sintering temperatures to fractures through grain interiors at higher sintering temperatures, suggesting that grain bonding is influenced by increasing sintering temperature. Additionally, samples sintered at 950 °C for shorter holding times (achieving relative densities of 98.2%, 95.4%, and 90.9%) exhibit visible pores on freshly fractured surfaces (Fig. S6). By reducing the holding time, pores ranging from nanometers to microns are successfully introduced, with both pore content and size increasing as relative density decreases. This tunable porosity, achieved through controlled processing, offers a unique strategy for tailoring microstructural features to enhance phonon scattering.

Subsequently, detailed phase analysis was conducted on samples sintered at 800, 850, 900, 950, 1000, and 1050 °C through powder XRD and Rietveld analyses (Fig. S7 and Table S2)[38]. The results show that the refined lattice parameters are consistent with those reported for similar (Zr,Hf)NiSn compositions (Table S3). Furthermore, as the sintering temperature increases, the weight percent ($wt\%$) of the half-Heusler phase increases (Fig. 2a), reaching ~96% for samples sintered at 850 °C and above. Minor oxide impurities, such as HfO$_2$ and (ZrHf)O$_2$, are detected, possibly due to residual feedstock metal or reactions with traces of oxygen during synthesis. After excluding these impurities, the occupancy of interstitial (4 d) sites was quantified. A positive correlation is found between sintering temperature and the amount of Ni interstitial defects, with 2% of interstitial Ni observed in samples sintered at 950 °C and above (Figs. 2a and S7; Table S2). To further investigate the Ni occupancies at the interstitial sites, transmission electron microscopy (TEM) and Cs-corrected high-angle annular dark-field scanning transmission electron microscopy (HAADF-STEM) were performed on the Zr$_{0.75}$Hf$_{0.25}$NiSn$_{0.99}$Sb$_{0.01}$ sintered at 950 °C. The selected area electron diffraction (SAED) patterns are well indexed to the F$-43m$ space group of the half-Heusler structure (Fig. 2b). HAADF-STEM image (Fig. 2c) clearly shows that 4 d sites are partially occupied by bright spots, directly visualizing interstitial Ni defects. The

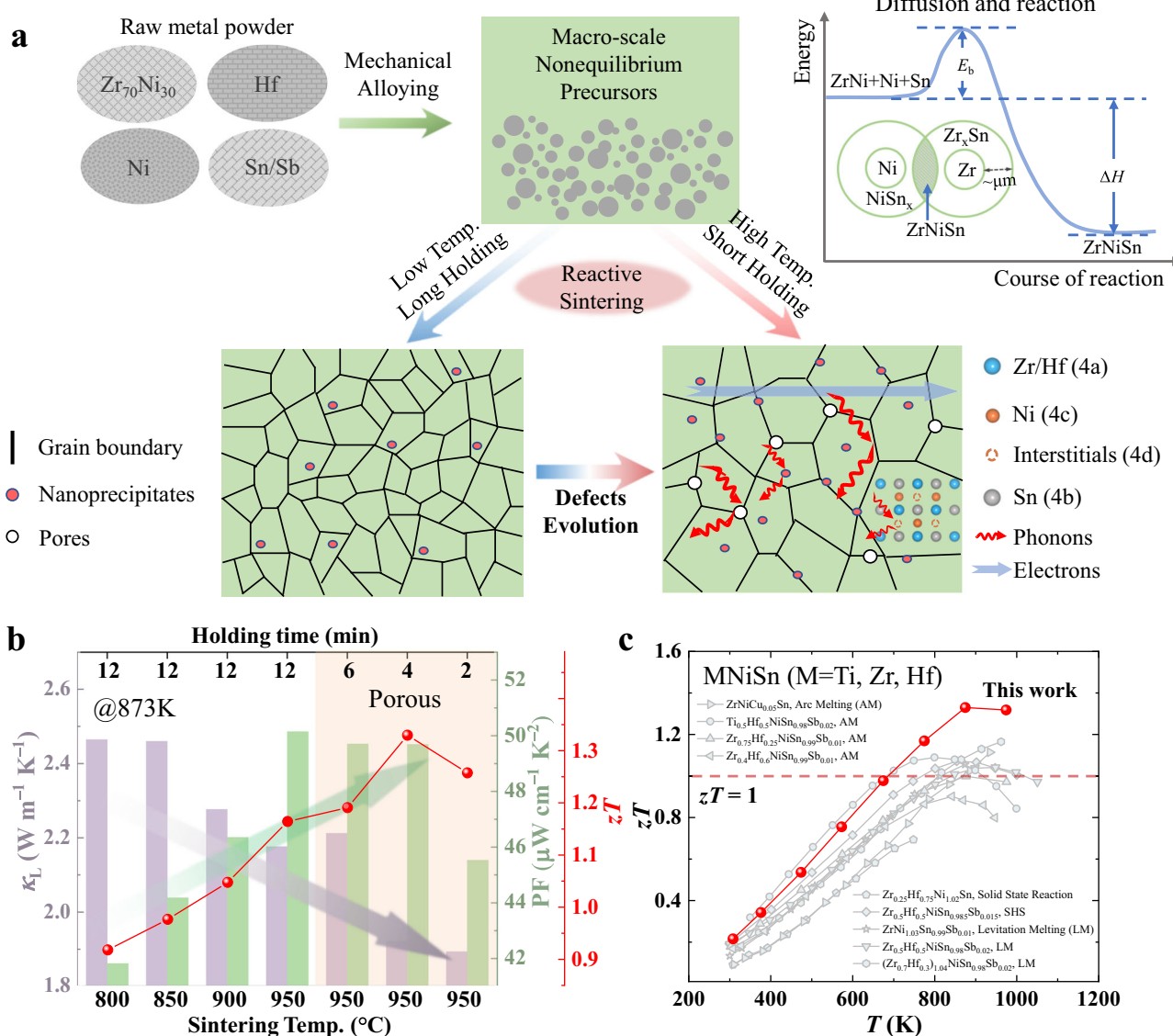

**Fig. 1 | Synthesis of ZrNiSn-based half-Heusler compounds with hierarchical structures through short-duration mechanical alloying and reactive sintering, leading to enhanced thermoelectric performance. a** Schematic of the reactive sintering process for ZrNiSn-based compounds and the development of hierarchical structures across multiple length scales during synthesis. **b** Lattice thermal conductivity ($\kappa_L$) and power factor (PF), and figure of merit ($zT$) of

$Zr_{0.75}Hf_{0.25}NiSn_{0.99}Sb_{0.01}$ synthesized under different sintering temperatures and holding times. **c** Comparison of the $zT$ value of $Zr_{0.75}Hf_{0.25}NiSn_{0.99}Sb_{0.01}$ with those of previously reported (Ti, Zr, Hf)NiSn-based materials synthesized using arc melting(AM), solid-state reaction, high-temperature self-propagation synthesis (SHS), and levitation melting (LM)[16,17,22,30–35].

interstitial Ni atoms contribute to local lattice distortions, which could influence both the thermal and electronic transport properties of the compounds.

Atom probe tomography (APT) was employed to analyze the structural features and the evolution of grain boundary (GB) composition with sintering temperature. Figure 2d shows the 3D point clouds of different distributions of elements and 1D composition profiles across the grain boundaries of samples sintered at 800 °C and 950 °C, labeled as GB1 and GB2, respectively. At 800 °C, a pronounced enrichment of Sn and deficiency of Ni is observed at GB1, accompanied by complementary Zr and Hf distributions. Note that the composition profiles for Zr and Hf are asymmetric relative to the position of the grain boundary. This observation typically indicates the solute-drag effect during the migration of grain boundaries[39]. It further indicates that grains in samples sintered at 800 °C exist in a thermodynamically non-equilibrium state, undergoing coarsening as the sintering temperature increases. In contrast, at 950 °C, the elements at GB2,

including Zr, Hf, Ni, Sn, and Sb, are nearly uniformly distributed. This indicates that the growth of grains and the interdiffusion of elements have been finished due to the high-temperature sintering. The corresponding 3D element distributions for these two samples are shown in Fig. S8. In addition, nanoprecipitates of 10–20 nm in size are observed in the matrix of both samples, consistent with the SEM images of fractured surfaces (Fig. S5). These clusters were identified as oxides of Hf and Zr through the 3D distribution mapping of elements and the composition proximity histogram (Fig. S9), aligning with the XRD refinement results indicating a total weight fraction of approximately 2% (Table S2). These nanoprecipitates could serve as additional phonon scattering centers, particularly for mid- to high-frequency phonons, thereby contributing to the reduction in thermal conductivity.

Due to the formation of a hierarchical structure, i.e., refined grain, interstitial defects, and nanoprecipitates, introduced by mechanical alloying and reactive sintering, the total thermal conductivity ($\kappa$) of dense $Zr_{0.75}Hf_{0.25}NiSn_{0.99}Sb_{0.01}$ samples range from 3.6 to

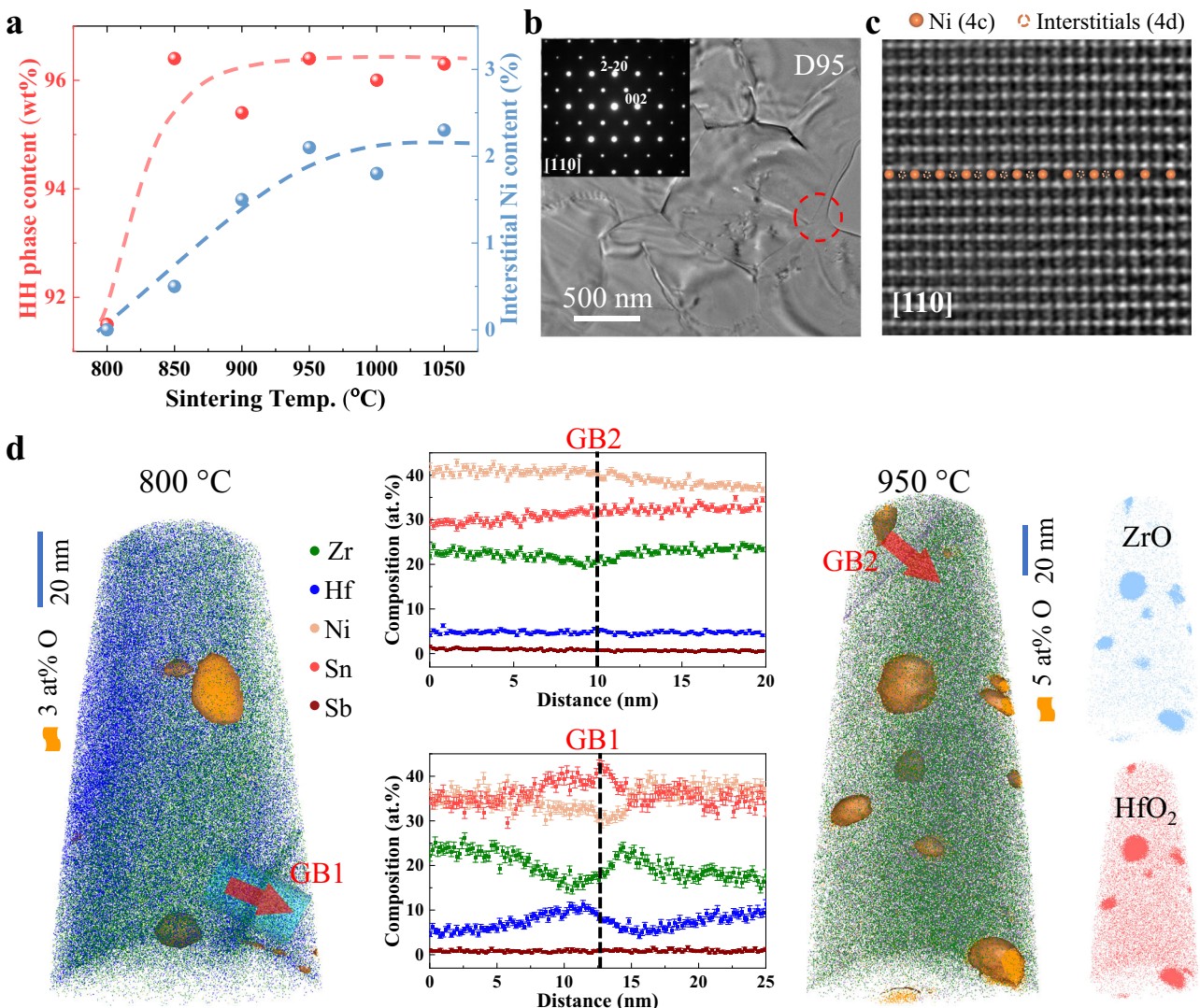

**Fig. 2 | Microstructure evolution depending on reactive sintering temperature. a** Content of the half-Heusler phase and 4d interstitial Ni occupancies in the dense $Zr_{0.75}Hf_{0.25}NiSn_{0.99}Sb_{0.01}$ samples. **b** Low-magnification TEM image and **c** Cs-corrected high-angle annular dark-field scanning transmission electron microscopy (HAADF-STEM) image of the porous $Zr_{0.75}Hf_{0.25}NiSn_{0.99}Sb_{0.01}$ samples sintered at 950 °C with a relative density of 95.4% (D95) taken along the [110] zone axis. The insert in (**b**) is the selected area of electron diffraction (SAED) of the region marked by a red dashed circle. The orange spheres and dashed circles in the HAADF-STEM image in (**c**) represent Ni atoms at the 4c and 4d positions, respectively. **d** Atom probe tomography (APT) reconstruction and corresponding 1D concentration profiles across the grain boundary (GB) in dense $Zr_{0.75}Hf_{0.25}NiSn_{0.99}Sb_{0.01}$ samples sintered at 800 °C and 950 °C, respectively. The corresponding 3D point clouds show Hf and Zr oxides (i.e., $HfO_2$ and ZrO) in the 950 °C samples.

4.4 W m$^{-1}$ K$^{-1}$ (Fig. 3a). These values are significantly lower compared to that of samples with the same composition prepared by arc melting (approximately 5.2 W m$^{-1}$ K$^{-1}$)[31]. To assess the impact of microstructure evolution on the lattice thermal conductivity ($\kappa_L$), we calculated $\kappa_L$ by subtracting the electronic thermal conductivity ($\kappa_e$), determined using the Wiedemann-Franz law, from $\kappa$. As shown in Fig. 3b, $\kappa_L$ at room temperature increases progressively with increasing sintering temperature, primarily due to grain growth and the elimination of secondary phases at boundaries (Fig. S5 and Fig. 2d), which reduces grain boundary density and improves acoustic matching, thus weakening phonon scattering. The sample with the smallest grain size, sintered at 800 °C, exhibits the lowest $\kappa_L$ at room temperature (3.2 W m$^{-1}$ K$^{-1}$). However, at 873 K, where bipolar diffusion occurs in all samples, $\kappa_L$ decreases significantly by approximately 10% with increasing sintering temperature (Fig. 3c). The sample sintered at 950 °C displays the lowest $\kappa_L$ of 2.1 W m$^{-1}$ K$^{-1}$. This apparent decrease in $\kappa_L$ is primarily attributed to the increased concentration of point defects (e.g.,

interstitial Ni defects), which effectively scatter high-frequency phonons.

To investigate the influence of interstitial Ni defects on phonon propagation, we performed first-principles calculations of the phonon dispersion. As shown in Fig. 3d, the phonon dispersion of $Zr_{27}Ni_{28}Sn_{27}$ (with a defect concentration of approximately 3.7%) shows the linewidths of phonon modes induced by the chemical disorder from Ni interstitial defects, which can increase the phonon scattering channels. Additionally, the introduction of interstitial defects softens the low-frequency acoustic phonons, resulting in a reduction of the calculated phonon group velocity in $Zr_{27}Ni_{28}Sn_{27}$ compared to ZrNiSn, particularly at approximately 2.5 THz (Fig. S10a). Further calculations reveal that increasing the concentration of interstitial Ni defects has a more pronounced effect on the phonon dispersion (Fig. S10b, c). These findings suggest that the increase in interstitial defects of ZrNiSn plays a crucial role in the reduction of lattice thermal conductivity at elevated temperatures.

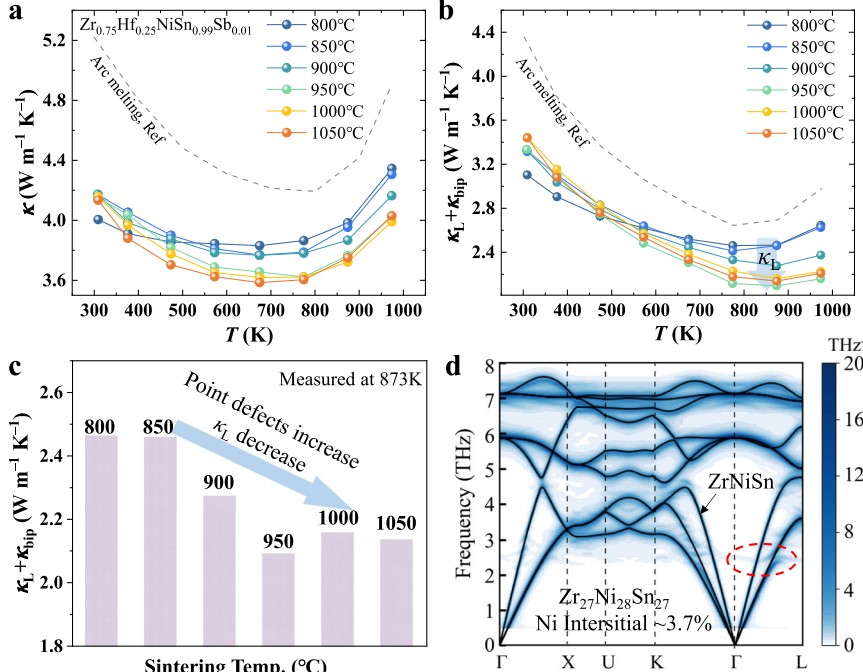

**Fig. 3 | Effect of reactive sintering temperature on thermal properties.**
**a** Temperature-dependent thermal conductivity ($\kappa$). Data for the sample prepared by arc melting is included for comparison[31]. **b** Temperature-dependent lattice and bipolar thermal conductivity ($\kappa_L + \kappa_{bip}$). **c** Comparison of $\kappa_L$ at 873 K for $Zr_{0.75}Hf_{0.25}NiSn_{0.99}Sb_{0.01}$ samples sintered at various temperatures. **d** Phonon dispersion of ZrNiSn (black solid line) and $Zr_{27}Ni_{28}Sn_{27}$ (blue color).

Samples sintered at 950 °C exhibited the lowest lattice thermal conductivity $\kappa_L$ of 2.1 W m⁻¹ K⁻¹. Building on this observation, we investigated the impact of holding time on the density and structure of the samples. By reducing the holding time, we were able to lower the density and create a porous structure, which led to an additional reduction in lattice thermal conductivity. As shown in Fig. 4a, $\kappa$ decreases significantly—by approximately 10% and 15% across the entire temperature range—for the porous samples with relative densities of 95.4% (D95) and 90.9% (D91), respectively, compared to the fully dense sample (D100). This decrease is primarily attributable to a marked reduction of $\kappa_L$ in the porous samples (Fig. 4b). Notably, the minimum $\kappa_L$ achieved is 1.9 W m⁻¹ K⁻¹ at 873 K for the D91 and D95 samples, which is among the lowest values for the state-of-the-art half-Heuslers[10].

To gain further insight into this phenomenon, we measured the room-temperature longitudinal and transverse sound velocities ($v_l$, $v_t$) of the samples. As shown in Fig. 4c, the resulting sound velocities ($v_s$) for the D95 and D91 samples are 3110.9 m/s and 3130.6 m/s, respectively—approximately 8% lower than that of the D100 sample (3379.2 m/s). The lower sound velocity in the porous samples is attributed to the fact that pores—whether vacuum- or gas-filled—cause the physical properties of the material to transition between those of the matrix material and the vacuum or gas phase[36,40]. Since sound velocity remains relatively constant in dense materials but drops dramatically in gases, the reduced sound velocity in porous samples directly contributes to lower lattice thermal conductivity. To quantify the effect of porosity on $\kappa_L$, we simulated the temperature-dependent $\kappa_L$ using the Debye model by incorporating the reduced sound velocity (Supplementary Information). As shown in Fig. 4d, the model predictions for both the fully dense D100 and the porous D95 samples align well with experimental data. This agreement further supports the conclusion that the reduction in thermal conductivity is driven by a synergistic effect, where phonon scattering from grain boundaries, point defects, and nanoprecipitates, alongside the pore-induced reduction in sound velocity, collectively contribute to the observed decrease in lattice thermal conductivity.

Since thermoelectric performance is intrinsically linked to electrical transport characteristics, we systematically measured the electrical conductivity ($\sigma$), room-temperature Hall concentration ($n_H$), and Hall mobility ($\mu_H$) of $Zr_{0.75}Hf_{0.25}NiSn_{0.99}Sb_{0.01}$ samples prepared at varying sintering temperatures (Fig. 5a and Fig. S11). Generally, the room-temperature $n_H$ for all samples is around $3 \times 10^{20}$ cm⁻³, while $\mu_H$ ranges from 25 to 30 cm² V⁻¹ s⁻¹ (Table S1 and Fig. S11), consistent with conventionally prepared samples[31,41]. The $\sigma$ decreases with increasing temperature, exhibiting degenerate semiconductor behavior. Notably, despite an increase in grain size at higher sintering temperatures, the $\sigma$ at room temperature decreases. This reduction is attributed to the elimination of metallic Sn-rich phases at grain boundaries (Fig. 2c), resulting in an enhancement of the grain boundary resistance. The temperature dependence of $\sigma$, with an exponential change from −0.4 to −0.2, aligns well with the mobility trend from −0.27 to −0.16 for samples sintered between 800 °C and 1050 °C (Fig. S12). This behavior suggests that at lower sintering temperatures (e.g., 800 °C), alloy scattering ($\mu \sim T^{-1/2}$) dominates, while at higher sintering temperatures, more significant scattering mechanisms come into play, possibly due to changes in grain boundary composition and an increase in interstitial Ni defects that correlate with the sintering temperature (Fig. 2a).

The Seebeck coefficient ($\alpha$) increases with higher sintering temperature (Fig. 5b), with bipolar diffusion becoming evident around 873 K, consistent with trends in lattice thermal conductivity. To further elucidate the transport properties, we analyzed the experimental data using the single parabolic band (SPB) model (Supplementary Information). The Pisarenko plot (Fig. 5c) at 300 K for $Zr_{0.75}Hf_{0.25}NiSn_{0.99}Sb_{0.01}$ samples yields an effective mass ($m^*$) of 3.4 $m_e$, in agreement with conventionally prepared samples[29]. Slight deviations observed in high-temperature sintered samples can be attributed to changes in the scattering mechanism. After adjusting the scattering factor ($\lambda$) from −0.5 to −0.3, which indicates stronger electron scattering, the data for samples sintered at higher temperatures align well with theoretical predictions. This increase in the scattering factor is consistent with the observed changes in electrical conductivity as a function of temperature.

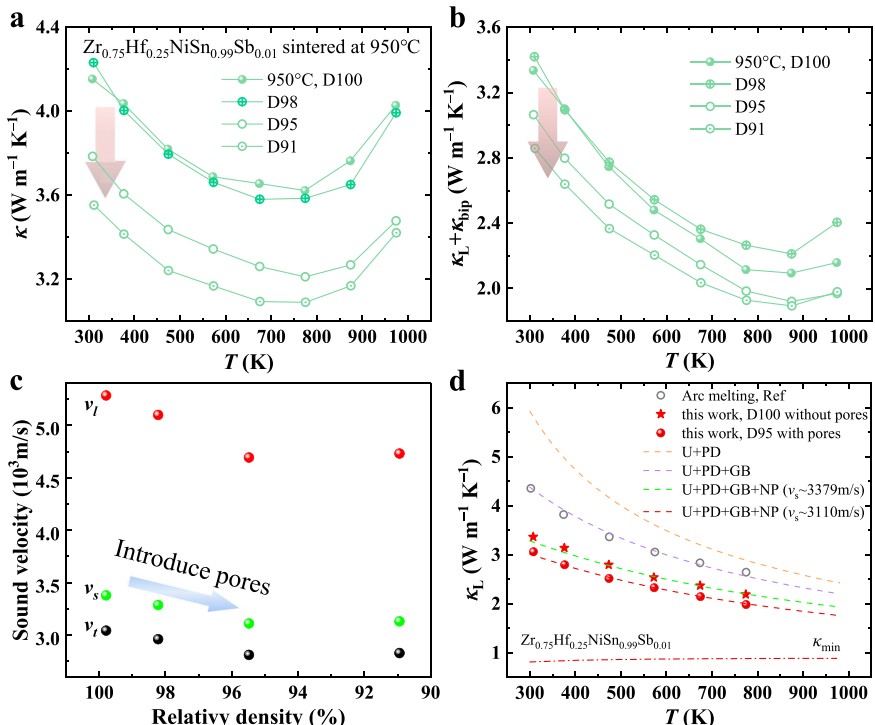

**Fig. 4 | Thermal properties of $Zr_{0.75}Hf_{0.25}NiSn_{0.99}Sb_{0.01}$ samples with relative densities ~ 99.8%, ~ 98.2%, ~ 95.4%, and ~ 90.9%, as labeled D100, D98, D95, and D91, respectively. a** Temperature-dependent $\kappa$. **b** Temperature-dependent $\kappa_L$. **c** Longitudinal and transverse sound velocities ($v_l$, $v_t$) and sound velocity ($v_s$) at room temperature. **d** Comparison of $\kappa_L$ between porous and non-porous samples.

The dash lines were calculated using the modified Debye model. U, PD, GB and NP represent the phonon–phonon Umklapp process, point defect, grain boundary, and nanoprecipitate scattering mechanisms for phonons, respectively. Data for the sample prepared by arc melting is included for comparison[31].

The calculated power factor (PF) is shown in Fig. 5d. With increasing sintering temperature, the PF improves over the entire temperature range. The detailed electrical transport properties of porous $Zr_{0.75}Hf_{0.25}NiSn_{0.99}Sb_{0.01}$ samples are presented in Fig. S13. Porous samples (e.g., D95) exhibit a higher Seebeck coefficient at elevated temperatures compared to dense counterparts, due to the energy filtering effects from pores and interfaces[16,42,43]. Consequently, a maximum PF of ~51 µW cm$^{-1}$ K$^{-2}$ is obtained at 873 K for the dense samples sintered at 950 °C and 1000 °C, and a PF of ~50 µW cm$^{-1}$ K$^{-2}$ for the sample with a relative density of 95.4% (D95). These values are comparable to the state-of-the-art $n$-type MNiSn materials synthesized using conventional methods such as arc melting, levitation melting, and high-temperature self-propagation synthesis[24,31,32,44]. This behavior demonstrates that the manipulation of multi-scale architectures−comprising interstitial defects, grain boundaries, nanoprecipitates, and pores− through controlled reactive sintering parameters does not degrade electron transport in $n$-type ZrNiSn-based compounds and thereby even maintains high electrical performance.

The simultaneous improvement of PF and reduction of $\kappa$ due to the microstructural and defect evolution results in an enhanced thermoelectric figure of merit ($zT$) for $Zr_{0.75}Hf_{0.25}NiSn_{0.99}Sb_{0.01}$ samples (Fig. 5e). The dense samples sintered at 950 °C and 1000 °C achieve $zT$ values as high as ~1.2 at 873 K. A maximum $zT$ of ~1.33 is attained at 873 K for porous ZrNiSn-based half-Heusler compounds, with good thermal stability and reproducibility (Figs. S14–S16), underscoring the advantages of porosity in decoupling electron and phonon transport. The average $zT$ ($zT_{average}$) of $Zr_{0.75}Hf_{0.25}NiSn_{0.99}Sb_{0.01}$ sample with a relative density of ~95.5% is calculated to be ~0.87 in the 310−973 K range and ~1.13 in the 573−973 K range, both of which exceed the state-of-the-art performance of $n$-type MNiSn compounds synthesized using conventional methods[16,17,30−32,34,35] (Fig. 5f).

## Discussion

In summary, we successfully synthesized high-performance n-type ZrNiSn-based half-Heusler thermoelectrics through a cost-effective, scalable mechanical alloying and reactive sintering method. The hierarchical microstructure, consisting of interstitial defects, grain boundaries, nanoprecipitates, and pores, was precisely controlled, enabling enhanced phonon scattering without compromising electron transport. This design led to a lattice thermal conductivity as low as 1.9 W m$^{-1}$ K$^{-1}$ and a maximum power factor of ~50 µW cm$^{-1}$ K$^{-2}$. The maximum $zT$ reached a remarkable value of 1.33 at 873 K, with an average $zT$ of 0.87, surpassing the performance of MNiSn compounds synthesized via conventional methods. Beyond ZrNiSn-based compounds, this reactive sintering strategy holds promise for other thermoelectric material systems, offering a scalable pathway to optimize microstructures and further enhance performance. Future efforts integrating advanced processing techniques for precise porosity control and optimized composition tuning could further enhance thermoelectric efficiency, accelerating the development of high-performance materials for practical energy conversion applications.

## Methods

### Sample preparation

The raw elements, including ZrNi powder (Zr:Ni; 70:30 wt%, 99.2%, Alfa Aesar), Hf powder (99.6%, Alfa Aesar), Ni powder (99.7%, Alfa Aesar), Sn powder (99.5%, Alfa Aesar), and Sb powder (99.5%, Alfa Aesar), were weighed according to the desired stoichiometry and loaded into ball milling jars. Each jar contained two stainless steel balls with a diameter of 12.7 mm. The process was conducted in an Ar-filled glovebox to maintain an inert atmosphere. Mechanical alloying was then conducted using a high-energy ball mill (SPEX 8000D) for 6 h, with the powder being loosened every 2 h. The precursor obtained from ball milling was then compressed into 10 mm diameter discs using a spark

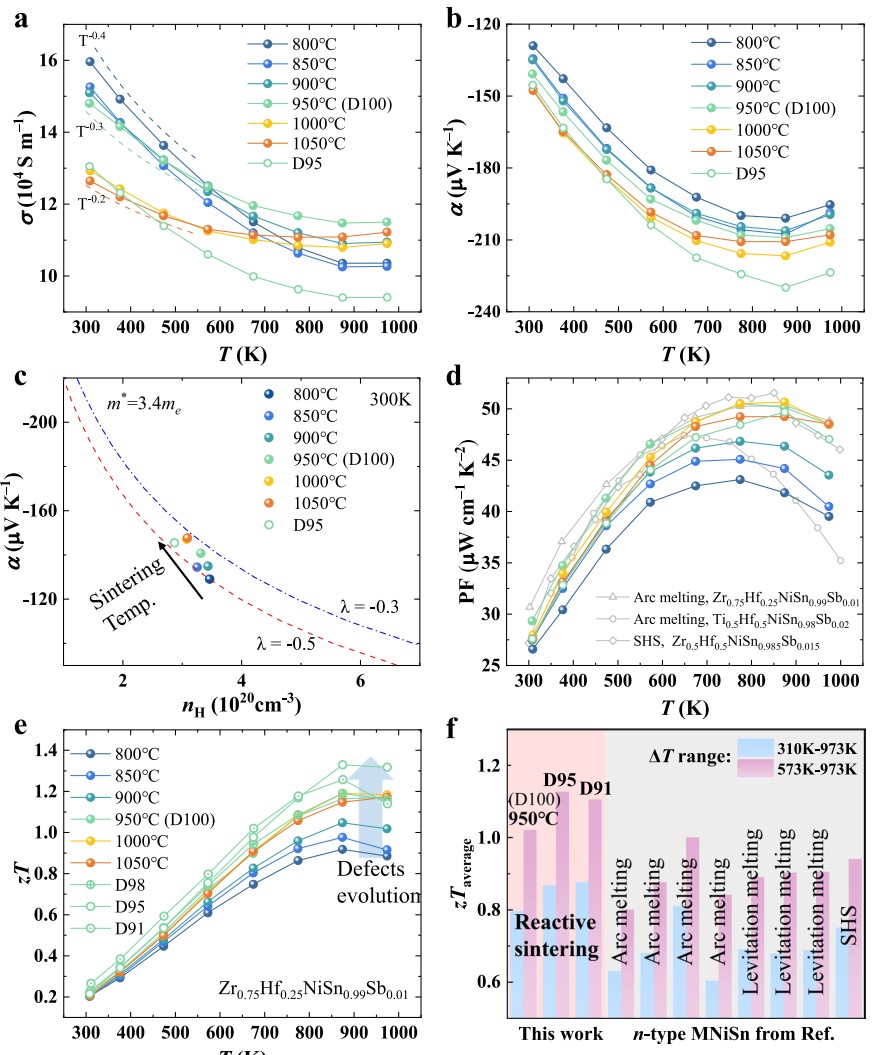

**Fig. 5 | Electrical transport properties and dimensionless figure of merit ($zT$).**
**a** Temperature-dependent electrical conductivity ($\sigma$). **b** Temperature-dependent
Seebeck coefficient ($\alpha$). **c** Pisarenko plot at 300 K. **d** Temperature-dependent power
factor (PF). Data for the samples prepared by arc melting and self-propagation

high-temperature synthesis (SHS) are included for comparison[24,31,32,44].
**e** Temperature-dependent $zT$. **f** Comparison of the average $zT$ ($zT_{average}$) of
our $Zr_{0.75}Hf_{0.25}NiSn_{0.99}Sb_{0.01}$ with those of previously reported MNiSn-based
compounds synthesized using conventional methods[16,17,30–32,34,35].

plasma sintering (SPS) apparatus (AGUS SPS-210Gx). Sintering was
carried out at a pressure of 50 MPa under vacuum conditions. The
sintering temperature ranged from 750 °C to 1100 °C, with 50 °C
intervals and a holding time of 12 min. To create compounds with a
porous structure, sintering was also performed at 950 °C with varying
holding times of 2, 4, and 6 min, respectively. The sintered samples
were subsequently cut and polished to the desired dimensions for
thermoelectric property measurements.

Reasons for using ZrNi alloy powder (Zr:Ni = 70:30 wt%), rather
than pure Zr, as raw material: (1) Pure zirconium powder is highly
reactive and typically packaged in water with organic stabilizers, which
can compromise purity; (2) Zr sponges or granules are hard and
require extensive ball milling or cryogenic grinding (e.g., tens of hours)
for synthesis, which may introduce impurities; (3) ZrNi alloy powder
provides high activity and purity, making it a more suitable and effi-
cient alternative to pure zirconium.

**Transport property measurement**
Electrical conductivity ($\sigma$) and Seebeck coefficient ($\alpha$) were
measured simultaneously using commercial equipment (LSR-3,
Linseis). Thermal conductivity was calculated by $\kappa = D\rho C_p$, where

thermal diffusivity ($D$) was measured using a laser-flash apparatus
(LFA1000, Linseis), specific heat ($C_p$) was obtained from ref. 45,
and mass density ($\rho$) was measured using the Archimedes prin-
ciple. Hall concentrations ($n_H$) were measured by a physical
property measurement system (Dynacool cryostat with a 9 T
magnet, Quantum Design) in the temperature range of 10 K to
room temperature, employing the Hall bar method under ±9 T
magnetic induction. The corresponding electrical conductivity
was measured by the thermal transport option (TTO) of the same
system. Hall mobility ($\mu_H$) was calculated by $\mu_H = \sigma/n_H e$, where $e$ is
the elementary charge. Thermogravimetric analysis (TGA) and
differential scanning calorimetry (DSC) were performed using a
METTLER TGA/DSC 3+ instrument with a 70 μL platinum crucible
under a nitrogen atmosphere, holding at 35 °C for 10 min, fol-
lowed by heating to 700 °C at 10 °C/min and a final 10 min iso-
thermal step. Room-temperature sound velocity was determined
using the RITEC Advanced Ultrasonic Measurement System RAM-
5000 within ±2% error. The measurement errors were 4%, 5%, and
6% for electrical conductivity, Seebeck coefficient, and thermal
conductivity, respectively. The uncertainties in thermal con-
ductivity were attributed to a 2% error in mass density and a 4%

error in thermal diffusivity. Consequently, the uncertainties in power factor and $zT$ were 10% and 15%, respectively. To enhance figure readability, the curves were plotted without error bars.

## Microstructure characterization

Powder diffraction patterns of the ball-milled precursor are measured on an Empyrean powder X-ray diffractometer (PANALYTICAL GmbH, Alpha-1 system), equipped with Cu-Kα$_1$ radiation (primary beam monochromator $\lambda = 1.54059$ Å). Bulk X-ray diffraction (XRD) analysis of all samples was performed on polished surfaces using a Bruker D8 Advance diffractometer with a Co Kα source to identify the phases. High-resolution X-ray powder diffraction patterns were collected using a STOE Stadi P diffractometer equipped with a Mo source, a Ge (111) primary beam monochromator, and a Mython 1 K detector (Dectris), yielding the $K_{\alpha 1}$ wavelength of 0.7093 Å in a flat sample transmission geometry. Rietveld analyses[38] of the XRD patterns were carried out using the FullProf software[46]. For the analyses, intensity, background, zero shift, lattice parameters, isotropic displacement parameter for each atom, peak asymmetries, $U$ from the Caglioti formula for strain, $Y$ for the crystallite size and the $4d$ site occupancy of Ni were included in the calculations. Other possibilities for $4d$ occupancies were checked but gave implausible results, like a negative occupancy. The $4d$ occupancy was refined individually, and the temperature factor was constrained with the $4c$ position. For all sample analyses, an instrumental resolution file was provided with the parameters taken from a Rietveld analysis of a NIST 640d silicon powder standard. Microstructure analysis was conducted using a field-emission scanning electron microscope (SEM, Zeiss Sigma 300). Elemental distribution was examined via energy-dispersive X-ray spectroscopy (EDX). The particle size of the ball-milled precursors and the grain size of bulk samples were determined from SEM images using ImageJ software. Transmission electron microscopy (TEM) and high-angle annular dark-field scanning transmission electron microscopy (HAADF-STEM) investigations were performed using a JEM-ARM 200F electron microscope, which is equipped with a cold field-emission gun (FEG) source and double-Cs correctors, operating at 200 kV. The needle-shaped atom probe tomography (APT) specimens were prepared by SEM-FIB dual beam focused ion beam (Helios NanoLab 650, FEI) following the standard "lift-out" method. APT measurements were conducted on a local electrode atom probe (LEAP 5000 XS, CAMECA). The laser pulses were adapted with a wavelength of 355 nm, a pulse duration of 10 ps, and a pulse energy of 10 pJ. A pulse repetition rate of 200 kHz with an average detection rate of 0.5%, an ion flight path of 100 mm, and a specimen base temperature of 50 K was utilized. The reconstruction of the 3D atom maps was performed using the visualization and analysis software package AP Suite 6.3.

## Theoretical calculations

The calculations are implemented using the Vienna Ab Initio simulation package (VASP) based on density functional theory (DFT)[47] with the projector augmented wave (PAW) method and Perdew-Burke-Ernzerhof (PBE) exchange-correlation functional[48]. The cut-off energy of the plane wave is set to 500 eV. The energy convergence value between two consecutive steps is set as $10^{-5}$ eV, and the maximum Hellmann-Feynman (HF) force acting on each atom is $10^{-3}$ eV/Å. The structure of ZrNiSn, Zr$_8$Ni$_9$Sn$_8$ and Zr$_{27}$Ni$_{28}$Sn$_{27}$ is optimized under $9 \times 9 \times 9$, $5 \times 5 \times 5$ and $3 \times 3 \times 3$ **k** mesh, respectively. The phonon properties are calculated by the finite displacement method with the Phonopy code[49]. The $3 \times 3 \times 3$, $2 \times 2 \times 2$ and $1 \times 1 \times 1$ supercells are used for the calculation of harmonic interatomic force constants (IFCs) of ZrNiSn, Zr$_8$Ni$_9$Sn$_8$ and Zr$_{27}$Ni$_{28}$Sn$_{27}$, respectively. The unfolding phonon dispersion of Zr$_8$Ni$_9$Sn$_8$ and Zr$_{27}$Ni$_{28}$Sn$_{27}$ is calculated by the UPHO code[50].

## Data availability

The data that support the findings of this study are available in Zenodo with the identifier(s) https://doi.org/10.5281/zenodo.15766977.

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

## Acknowledgements

R.H. acknowledges financial support from Deutsche Forschungsgemeinschaft (DFG), Project Number 453261231, as well as from the European Union's Horizon Europe research and innovation program (ERC Starting Grant, TENTATION, 101116340). Y. Wang acknowledges financial support from the National Science Foundation of China (Grant Nos. 12374021,12074409). Y. Wu acknowledges financial support from the National Science Foundation of China (Grant No. 12304038). Thanks to Mr. Kai Xu from Donghua University for his support on the TGA and DSC.

## Author contributions

X.A., K.N., and R.H. conceived the ideas and designed the work. X.A., L.G., W.X., Y. Wang, and A.S. carried out the experiments, including material preparation and characterization, XRD refinement, and TEM characterization. H.L. and Y.Y. contributed to APT measurements. Y. Wu conducted the DFT calculations. X.A. and Q.Z. wrote the draft. G.J.S., Y.Y., D.M., K.N., and R.H. contributed to the discussion and editing. All authors reviewed and commented on the manuscript.

## Funding

## Competing interests

The authors declare no competing interests.
