## [Transparent Peer Review file · Nature Communications]

High-Performance ZrNiSn-based Half-Heusler Thermoelectrics with Hierarchical Architectures Enabled by Reactive Sintering

Corresponding Author: Dr Ran He

Version 0:

Reviewer comments:

Reviewer #1

(Remarks to the Author)

The manuscript presents a study on microstructural engineering to manipulate ZrNiSn-based half-Heusler thermoelectric materials. By introducing a hierarchical microstructure—comprising interstitial defects, grain boundaries, nanoprecipitates, and pores—the authors aim to enhance phonon scattering, thereby reducing lattice thermal conductivity. A peak zT value of 1.33 at 873 K is reported, which is indeed notable. The manuscript provides a detailed discussion and explanation of the microstructural features and their influence on transport properties, as well as highlighting performance advancements. However, several critical issues need to be addressed to strengthen the manuscript and clarify the novelty and underlying mechanisms.

1. The title and abstract refer to a “novel reaction sintering approach,” yet the methodology described—ball milling followed by spark plasma sintering (SPS)—is one of the most commonly used preparation techniques in thermoelectric research, including for half-Heusler materials. As such, this claim may seem misleading. So I am not sure what the novelty of the preparation approach here is, nor why it should be highlighted.
2. The temperature dependence of σ has been considered as alloy scattering dominance in the sample (800 °C), but what is the reason for the change after increasing the sintering temperature. The increased grain size should have an opposite effect here.
3. The reported higher lattice thermal conductivity at room temperature in samples sintered at higher temperatures is counterintuitive, given the introduction of multiple defects designed to scatter phonons across all temperatures.
4. The D95 samples exhibit a significantly lower carrier concentration, which appears to play an important role in optimizing the electrical transport properties. Could the authors clarify the underlying reason for this reduction in carrier concentration?
5. The introduction of pores can enhance phonon scattering but may compromise the mechanical and thermal stability of the material. The manuscript should include an evaluation of the thermal stability of the sample, possibly through thermal cycling tests or long-term stability assessments, to ensure the material's reliability in practical applications.

Reviewer #2

(Remarks to the Author)

Ai et al. present an interesting study on the optimization of $\text{Zr}_{0.75}\text{Hf}_{0.25}\text{NiSn}_{0.99}\text{Sb}_{0.01}$ alloy through reactive sintering, achieving a dimensionless figure of merit (zT) of 1.33 at 873 K. The reported low lattice thermal conductivity ($\sim 1.9 \text{ W m}^{-1}\text{K}^{-1}$) and high power factor ($50 \mu\text{W cm}^{-1}\text{K}^{-2}$) underscore the potential of reactive sintering for enhancing thermoelectric performance. However, the presence of multiple phases and associated challenges in regulating them, highlight critical shortcomings in both the processing method and the consistency of the findings. Thus, several inconsistencies, outlined below, must be addressed through major revision.

1. Figure S2 shows that the alloyed $\text{Zr}_{0.75}\text{Hf}_{0.25}\text{NiSn}_{0.99}\text{Sb}_{0.01}$ contains numerous defects, although after spark plasma sintering it forms a half-Heusler phase, as shown in Figure S3. However, a comparison with standard ZrNiSn powder diffraction patterns reveal significant differences in the relative peak intensities. Why? Additionally, it would be helpful to understand how much the lattice parameter varies to values reported for similar compositions in previous studies.

2. The backscattered electron images in Figure S4 show Zr-rich regions for samples processed at $T > 850\text{ }^{\circ}\text{C}$, while ZrNi appears to dominate at lower temperatures. However, the underlying reason for this temperature-dependent phase distribution is not well explained. Could the authors clarify the mechanism driving this change? Such as clarifying differences in phase stability, diffusion kinetics, or other factors?
3. The phases identified in the sample include $\text{Zr}_{0.75}\text{Hf}_{0.25}\text{NiSn}_{0.99}\text{Sb}_{0.01}$, HfO_2 , Ni_3Sn_4 , Ni_3Sn_2 , and ZrHfO_2 . The proportions of these phases are crucial as they can significantly impact on the properties of the Half-Heusler alloy, which is widely recognized for high sensitivity to composition and stoichiometry. Moreover, how is reproducibility ensured in the synthesis of these samples? Also, which specific phase contributes to a reduction in lattice thermal conductivity?
4. Are the synthesized samples thermally stable within the measurement temperature ranges? Please provide clarification by including the heating and cooling curves. Additionally, temperature-dependent heat capacity measurements would be valuable for assessing the phase behavior within the measured/ sintering temperature range.
5. Comment on the measurement error in sound velocity. Similarly, discuss the propagation errors and their impact on the accuracy of the ZT measurements for the synthesized alloys.
6. The authors have apparently overlooked recent studies on high-performance ZrNiSn half-Heusler alloys, where a $zT > 1$ was achieved through various doping strategies, particularly with Hf substitution and role of Ni interstitials. Such omission weakens the comparison between the synthesized samples and the current state of the art.
7. There are prevailing typos and errors which need to be thoroughly checked and corrected. For instance, "Piont defects decrease" in Figure 3c. Unit of temperature should be in Kelvin.

Reviewer #3

(Remarks to the Author)

Comments to the authors (Manuscript id: NCOMMS-25-26960)

This manuscript presents a novel approach to synthesize ZrNiSn-based half-Heusler compounds with multi-scale architectures comprising interstitial defects, grain boundaries, nanoprecipitates, and pores, enabling strong phonon scattering. Thus, significantly low lattice thermal conductivity $1.9\text{ W m}^{-1}\text{ K}^{-1}$ and a high power factor of $50\text{ }\mu\text{W cm}^{-1}\text{ K}^{-2}$ leading to a notably high thermoelectric figure of merit (zT) = 1.33 at 873 K is reported in $\text{Zr}_{0.75}\text{Hf}_{0.25}\text{NiSn}_{0.99}\text{Sb}_{0.01}$. A thorough investigation on microstructure and its impact on thermoelectric properties is provided by the authors. The results are well supported by experimental as well as theoretical investigations. This work may guide the future research for achieving low thermal conductivity and optimizing thermoelectric performance in half-Heusler compounds. Therefore, I recommend this manuscript to be published in Nature Communications after minor revision.

1. Fig. S8; why are Sn and Sb distributed uniformly in oxide regions like the matrix?
2. Page #9, line, "However, at 873 K, where dipolar diffusion occurs in all samples...". Please correct typo, dipolar should be bipolar.
3. Page #10, paragraph, "To investigate the influence...". Please check and correct figure numbers (Figure S10a-d) and (Figure S10e and f).
4. Page #13, line, "The temperature dependence ...from -0.27 to -0.16 between 10 K and 300 K (Figure S12)." It should be the sintering temperature instead of measurement temperature (10 K and 300 K), please check and correct it.
5. Fig. 5b and S13b; why do porous samples (such as D95) show a sharp rise in the Seebeck coefficient at high temperature compared to high density samples? This is compensating for the electrical conductivity reduced due to porosity and thus leading to a high-power factor as well as high zT . Therefore, an explanation is needed in the manuscript.

Version 1:

Reviewer comments:

Reviewer #1

(Remarks to the Author)

All issues have been adequately addressed. I recommend acceptance of the manuscript.

Reviewer #2

(Remarks to the Author)

The revised manuscript satisfactorily addresses many of the previous comments. However, Figure 5f, which compares the average thermoelectric figure of merit (zT_{avg}) of $\text{Zr}_{0.75}\text{Hf}_{0.25}\text{NiSn}_{0.99}\text{Sb}_{0.01}$ with other MNiSn-based compounds, overlooks several recent studies that report zT_{avg} values exceeding 1 using conventional synthesis routes. Such omissions limit the impact and relevance of comparative analysis, and it is encouraged to incorporate more recent and higher-performing data points to provide a more comprehensive perspective.

Reviewer #3

(Remarks to the Author)

The authors have responded to all of my previous comments in a satisfactory manner. The revised manuscript is clear, well-structured, and scientifically robust. I find no remaining issues and recommend the manuscript for acceptance in its current form.

Dear Reviewers,

We sincerely thank you for your time and effort in reviewing our manuscript entitled “*High-Performance ZrNiSn-based Half-Heusler Thermoelectrics with Hierarchical Architectures Enabled by Reactive Sintering*” (NCOMMS-25-26960). We greatly appreciate your constructive comments and valuable suggestions, which have helped us improve the quality and clarity of our work. In response, we have carefully revised the manuscript and supplementary information. For clarity, your comments are presented in **blue**, followed by our point-by-point responses in **black**. All revisions made to the manuscript and supplementary information are highlighted in **red** in the revised files.

Please find our detailed responses below.

Reviewer #1 (Remarks to the Author):

The manuscript presents a study on microstructural engineering to manipulate ZrNiSn-based half-Heusler thermoelectric materials. By introducing a hierarchical microstructure—comprising interstitial defects, grain boundaries, nanoprecipitates, and pores—the authors aim to enhance phonon scattering, thereby reducing lattice thermal conductivity. A peak zT value of 1.33 at 873 K is reported, which is indeed notable. The manuscript provides a detailed discussion and explanation of the microstructural features and their influence on transport properties, as well as highlighting performance advancements. However, several critical issues need to be addressed to strengthen the manuscript and clarify the novelty and underlying mechanisms.

Response:

We sincerely appreciate your thorough evaluation and constructive feedback. We agree that addressing the raised concerns is essential to clarify the novelty further and reinforce the mechanistic understanding of our work. Accordingly, we have revised the manuscript to enhance its clarity, scientific depth, and overall presentation. Detailed point-by-point responses are provided below.

1. The title and abstract refer to a “novel reaction sintering approach,” yet the methodology described—ball milling followed by spark plasma sintering (SPS)—is one of the most commonly used preparation techniques in thermoelectric research, including for half-

Heusler materials. As such, this claim may seem misleading. So, I am not sure what the novelty of the preparation approach here is, nor why it should be highlighted.

Response:

Thank you for this insightful comment. We fully agree that the combination of ball milling and spark plasma sintering (SPS) is a widely used processing route in thermoelectric research, including for half-Heusler compounds. The novelty of our work does not lie in the use of SPS *per se*, but rather **in the implementation and optimization of a reactive sintering strategy**—starting from ZrNi alloy and elemental powders of Hf, Ni, Sn, and Sb—for the synthesis of ZrNiSn-based half-Heusler compounds. This approach distinctly differs from conventional melting-based synthesis methods, as detailed in Table R1.

Table R1. Comparison of preparation process and zT_{\max} of ZrNiSn-based half-Heusler compounds.

Raw materials	Preparation process	zT_{\max}	Ref.
Hf piece, Zr slug, Ni slug, Sn wire and Sb shot	Induction Melting (several times) + Ball Milling (4 hours) + SPS	0.9	[1]
Zr piece, Ni rods, Sn block, and Sb block	Levitation Melting (6 times) + Ball Milling (1 h) + SPS	1.05	[2]
Hf slice, Zr ingot, Ni ingot, Sn granule, and Sb ingot	Levitation Melting (3 times) + Ball Milling (30 min) + SPS (+Annealing 15 days)	1.2	[3]
Hf, Zr, Ni, Sn, and Sb	Arc Melted + Ball Milling (5–12 hours) + Hot Pressing	1	[4]
Hf, Zr, Ni, Sn, and Sb powder	Self-Propagating High-Temperature Synthesis (SHS) + Ball Milling (1 hour) + Hot Press (+Annealing)	1.1	[5]
Hf, Zr₇₀Ni₃₀ , Ni, Sn, and Sb powder	Ball Milling (6 hours) + SPS	1.33	This work

- [1] *Mater. Today*, 2020, 36, 63.
 [2] *Nat. Commun.*, 2024, 15, 5978.
 [3] *Adv. Energy Mater.*, 2024, 2402399.
 [4] *Adv. Energy Mater.*, 2013, 3, 1210.
 [5] *Energy Environ. Sci.*, 2019, 12, 3390.

In the majority of prior studies, as summarized in **Table R1**, ball milling and SPS are typically employed **after** an initial melting step (*e.g.*, induction, arc, or levitation melting). These melting processes yield near-phase-pure ingots, which are subsequently milled to reduce grain size and improve sinterability before final densification via SPS. In contrast, our method **entirely bypasses** the melting step. Instead, we use ball milling as a solid-state alloying technique to produce non-equilibrium precursors directly from elemental and intermetallic ZrNi powders (overcoming the low reactivity of the high hardness Zr granules or sponges as raw materials). These precursors are then subjected to SPS, during which both phase formation and densification occur concurrently—a hallmark of reactive sintering.

While there have been some efforts to synthesize ZrNiSn-based half-Heuslers using only ball milling and SPS (*e.g.*, *ACS Appl. Mater. Interfaces*, 2021, 13, 38561–38568; *J. Solid State Chem.*, 2020, 285, 121203), such approaches often result in multiphase products with limited thermoelectric performance (typically peak $zT < 0.7$). Our work addresses this limitation by using ZrNi alloy as a key starting material, which enables a shorter milling time (6 hours) and a more controlled reaction path during SPS. This strategy yields a single-phase ZrNiSn-based compound with a peak zT of 1.33 at 873 K—the highest reported for a non-melted ZrNiSn-based half-Heusler, to the best of our knowledge.

In addition to avoiding energy-intensive melting steps, our approach offers improved scalability, reduced cost, and enhanced control over microstructure—all of which are critical for advancing the practical deployment of thermoelectric materials.

Accordingly, we have revised the manuscript to more accurately reflect the novelty of our work as the application and optimization of a reactive sintering route specific to the ZrNiSn system, rather than suggesting a generally new synthesis method.

2. The temperature dependence of σ has been considered as alloy scattering dominance in the sample (800 °C), but what is the reason for the change after increasing the sintering temperature. The increased grain size should have an opposite effect here.

Response:

Thank you for your thoughtful comment. You are absolutely right that increasing the sintering temperature (*e.g.*, from 800 °C to 950 °C) typically promotes grain growth, which would be expected to reduce grain boundary scattering and thus enhance electrical conductivity (σ). However, in our study, we observe a **decrease** in room-temperature σ with increasing sintering temperature—a seemingly counterintuitive result.

This behavior is primarily attributed to the evolution of secondary phases at the grain boundaries. Specifically, in samples sintered at lower temperatures (800 °C), we observe the presence of Sn-rich metallic phases at the grain boundaries (**Figure R1**). These secondary phases act as conductive bridges between grains, significantly reducing grain boundary resistance and thereby enhancing σ .

At higher sintering temperatures, these Sn-rich phases are eliminated, likely due to homogenization and diffusion-driven dissolution into the matrix. As a result, the low-resistance bridges are removed, leading to an **increase** in grain boundary resistivity. This effect outweighs the positive influence of grain growth, resulting in a net reduction in σ .

Figure R1. Atom probe tomography (APT) reconstruction and corresponding 1D concentration profiles across the grain boundary (GB) in dense $\text{Zr}_{0.75}\text{Hf}_{0.25}\text{NiSn}_{0.99}\text{Sb}_{0.01}$ samples sintered at 800 °C and 950 °C, respectively.

We have clarified this point in the revised manuscript (**Results section, page 14**) and added a supplementary discussion to explain the competing effects of grain size and grain boundary phase evolution on σ .

3. The reported higher lattice thermal conductivity at room temperature in samples sintered at higher temperatures is counterintuitive, given the introduction of multiple defects designed to scatter phonons across all temperatures.

Response:

Thank you for your valuable comment. We agree that the observed increase in lattice thermal conductivity (κ_L) at room temperature for samples sintered at higher temperatures appears counterintuitive, especially given that higher sintering temperatures are associated with the introduction of defects designed to enhance phonon scattering.

However, this behavior reflects the complex interplay of multiple microstructural factors that evolve with sintering temperature, each influencing κ_L in different and sometimes competing ways. For example:

- **Grain growth:** Higher sintering temperatures promote grain coarsening, which reduces grain boundary density and thus decreases phonon scattering at grain boundaries—*increasing* κ_L .
- **Point defects (interstitial Ni):** These tend to increase with higher sintering temperatures and act as phonon scatterers—*decreasing* κ_L .
- **Grain boundary modification:** Reduced segregation at higher sintering temperatures lessens the acoustic mismatch at grain boundaries—*increasing* κ_L .

The net effect on κ_L is therefore the result of a delicate balance among these opposing contributions. Importantly, while higher-temperature sintering introduces phonon-scattering features such as point defects and nanoprecipitates, their scattering effectiveness tends to become more significant at elevated measurement temperatures. **At room temperature, phonon transport is more strongly governed by grain boundary scattering.** Thus, the reduction in grain boundary density and the improved acoustic matching (due to the elimination of secondary phases at boundaries) in the high-temperature-sintered samples results in **increased** κ_L at low temperatures.

We have clarified this point in the revised manuscript (**Results section, pages 9 and 10**) and expanded the discussion to reflect the competing microstructural effects on phonon transport.

4. The D95 samples exhibit a significantly lower carrier concentration, which appears to play an important role in optimizing the electrical transport properties. Could the authors clarify the underlying reason for this reduction in carrier concentration?

Response:

Thank you for your comment. We would like to clarify that the Hall carrier concentration (n_H) of the D95 sample is $2.9 \times 10^{20} \text{ cm}^{-3}$, comparable to that of the other samples with values around $3 \times 10^{20} \text{ cm}^{-3}$ (**Figure R2**). This consistency reflects the uniform chemical composition across

all samples. We have revised the manuscript to make this point clearer and avoid potential misunderstanding (see **Results section, page 14, and Supplementary information Table S1**).

Figure R2. Hall carrier concentration (n_H) of all samples.

5. The introduction of pores can enhance phonon scattering but may compromise the mechanical and thermal stability of the material. The manuscript should include an evaluation of the thermal stability of the sample, possibly through thermal cycling tests or long-term stability assessments, to ensure the material's reliability in practical applications.

Response:

Thank you for raising this important point. To evaluate thermal stability, we conducted thermogravimetric analysis (TGA) and repeated electrical property measurements on the porous sample (D95). The TGA results (**Figure R3**) show no measurable mass change during six consecutive heating–cooling cycles from 150 °C to 700 °C, indicating good thermal stability. Further confirmation comes from the electrical conductivity (σ) and Seebeck coefficient (α) measurements (**Figure R4**): the five repeated curves coincide with each other across the entire temperature range (300–973 K), underscoring the material's stability under thermal cycling.

Figure R3. TGA results of six heating-cooling cycles from 150 °C to 700 °C of porous sample (D95).

Figure R4. Repeated measurements of (a) electrical conductivity (σ) and (b) Seebeck coefficient (α) of the porous sample (D95).

We have added these results to the revised manuscript (**Results section, page 15**) and included Figure R3 in the revised Supplementary Information (**Figure S14 and Figure S15**).

Reviewer #2 (Remarks to the Author):

Ai et al. present an interesting study on the optimization of $Zr_{0.75}Hf_{0.25}NiSn_{0.99}Sb_{0.01}$ alloy through reactive sintering, achieving a dimensionless figure of merit (zT) of 1.33 at 873 K. The reported low lattice thermal conductivity ($\sim 1.9 \text{ W m}^{-1} \text{ K}^{-1}$) and high power factor ($50 \mu\text{W cm}^{-1} \text{ K}^{-2}$) underscore the potential of reactive sintering for enhancing thermoelectric performance. However, the presence of multiple phases and associated challenges in regulating them, highlight critical shortcomings in both the processing method and the consistency of the

findings. Thus, several inconsistencies, outlined below, must be addressed through major revision.

Response:

We sincerely thank you for the careful assessment of our work and the constructive comments provided. We appreciate the recognition of the significance of our findings and the concerns raised regarding phase control and data consistency. These points have helped us critically re-examine and strengthen our manuscript. In the following, we provide detailed, point-by-point responses to your comments. The manuscript has been thoroughly revised to address all concerns and to improve clarity and scientific rigor.

1. Figure S2 shows that the alloyed $\text{Zr}_{0.75}\text{Hf}_{0.25}\text{NiSn}_{0.99}\text{Sb}_{0.01}$ contains numerous defects, although after spark plasma sintering it forms a half-Heusler phase, as shown in Figure S3. However, a comparison with standard ZrNiSn powder diffraction patterns reveal significant differences in the relative peak intensities. Why? Additionally, it would be helpful to understand how much the lattice parameter varies to values reported for similar compositions in previous studies.

Response:

Thank you for this important observation and apologize for the confusion caused. The apparent differences in the relative peak intensities in Figure S3 compared to standard ZrNiSn diffraction patterns were due to the non-uniform y-axis scaling during plotting. We have reprocessed the XRD data using a consistent intensity scale, and the corrected pattern is now provided in **Figure R5**. The revised plot confirms the formation of a well-crystallized half-Heusler phase with no abnormal deviations in relative peak intensities.

Figure R5. XRD patterns of (a) $\text{Zr}_{0.75}\text{Hf}_{0.25}\text{NiSn}_{0.99}\text{Sb}_{0.01}$ precursors prepared by mechanical alloying and (b) sintered $\text{Zr}_{0.75}\text{Hf}_{0.25}\text{NiSn}_{0.99}\text{Sb}_{0.01}$ bulk samples.

Additionally, we have compared the lattice parameters obtained in this work with those reported for similar (Zr,Hf)NiSn compositions. As shown in **Table R2** below, literature values typically range from 6.094 to 6.113 Å, while our calculated values fall within 6.097 to 6.103 Å, demonstrating good agreement.

Table R2. Comparison of lattice parameters obtained in this work with literature-reported values for (Zr,Hf)NiSn-based half-Heusler compounds.

Composition	Crystalline Form	Lattice Parameter (Å)	Sources
$\text{Zr}_{0.75}\text{Hf}_{0.25}\text{NiSn}_{0.99}\text{Sb}_{0.01}$	Polycrystalline	6.097 to 6.103	This work
ZrNiSn	Polycrystalline	6.113	Ref [1]
ZrNiSn	Polycrystalline	6.099	Ref [2]
ZrNiSn	Polycrystalline	6.106	Ref [3]
$\text{Hf}_{0.6}\text{Zr}_{0.4}\text{NiSn}_{0.98}\text{Sb}_{0.02}$	Polycrystalline	6.094	Ref [4]
$\text{Hf}_{0.25}\text{Zr}_{0.75}\text{NiSn}_{0.99}\text{Sb}_{0.01}$	Polycrystalline	6.098	Ref [5]
$\text{ZrNi}_{1.03}\text{Sn}_{0.99}\text{Sb}_{0.01}$	Polycrystalline	6.1	Ref [6]
ZrNiSn	Polycrystalline	6.109	Ref [7]
ZrNiSn	Single Crystal	6.1033	Ref [8]

[1] *Metall Trans*, **1970**, 1, 3159.

[2] *Ukrainskij Fizicheskij Zhurnal*, **1986**, 31, 1258.

[3] *Mater. Today Phys.*, **2023**, 33, 101049.

- [4] *J. Appl. Phys.*, **2010**, 108, 6.
- [5] *Adv. Energy Mater.*, **2013**, 3, 1210.
- [6] *Nat. Commun.*, **2024**, 15, 5978.
- [7] *Nano Energy*, 2020, 78, 105372.
- [8] *Adv. Sci.*, **2020**, 7, 1902409.

These results, along with the updated figure and discussion have been incorporated into the revised manuscript and Supplementary Information (**Results section, page 7 and Supplementary information Table S3**).

2. The backscattered electron images in Figure S4 show Zr-rich regions for samples processed at $T > 850\text{ }^{\circ}\text{C}$, while ZrNi appears to dominate at lower temperatures. However, the underlying reason for this temperature-dependent phase distribution is not well explained. Could the authors clarify the mechanism driving this change? Such as clarifying differences in phase stability, diffusion kinetics, or other factors?

Response:

Thank you for your thoughtful and important question. The temperature-dependent phase distribution observed in Figure S4 is indeed a central aspect of our study and can be attributed to a combination of thermodynamic and kinetic factors that govern phase evolution during reactive sintering.

Thermodynamically, the transformation of ZrNi into the desired half-Heusler phase involves overcoming a significant energy barrier. At lower sintering temperatures ($\leq 850\text{ }^{\circ}\text{C}$), the available thermal energy is insufficient to fully drive this transformation, resulting in the retention of residual ZrNi intermediate phases. The presence of such intermediates is common in rapid solid-state reactions, where limited diffusion prevents complete conversion. Similar behavior has been reported in other systems, such as Ni–Sn intermediates in half-Heusler compounds and Cu–Se intermediates in Cu_2Se systems (*e.g.*, *J. Mater. Chem. A*, 2018, 6, 19470–19478; *Nat. Commun.*, 2014, 5, 4908).

As the sintering temperature increases above $850\text{ }^{\circ}\text{C}$, enhanced atomic mobility enables more complete reactions among ZrNi, Sn, and other constituent elements, leading to a significant reduction in residual ZrNi phases. However, this progression is accompanied by the appearance

of Zr-rich half-Heusler domains. This can be understood as a local thermodynamic preference for forming stable Zr-rich variants in regions where unreacted or partially reacted ZrNi persists, especially under non-equilibrium conditions.

Kinetically, even at higher temperatures, the short sintering duration and the solid-state nature of the process can limit atomic diffusion and hinder the full homogenization of the elements. As a result, local composition inhomogeneities—particularly Zr-enriched regions—may remain and subsequently evolve into Zr-rich half-Heusler phases during the sintering process.

In summary, while thermodynamic factors explain the disappearance of ZrNi and the tendency toward half-Heusler phase formation, kinetic constraints account for the incomplete homogenization that gives rise to these Zr-rich domains. Together, these mechanisms explain the observed shift from ZrNi-rich microstructures at lower sintering temperatures to Zr-rich half-Heusler phases at higher temperatures.

We have clarified this mechanism in the revised manuscript (**Results section, page 6**) and updated the discussion accordingly.

3. The phases identified in the sample include $Zr_{0.75}Hf_{0.25}NiSn_{0.99}Sb_{0.01}$, HfO_2 , Ni_3Sn_4 , Ni_3Sn_2 , and $ZrHfO_2$. The proportions of these phases are crucial as they can significantly impact on the properties of the Half-Heusler alloy, which is widely recognized for high sensitivity to composition and stoichiometry. Moreover, how is reproducibility ensured in the synthesis of these samples? Also, which specific phase contributes to a reduction in lattice thermal conductivity?

Response:

Thank you for raising these important points. To ensure **reproducibility**, we employed a fixed synthesis protocol using commercially sourced ZrNi alloy and Hf, Ni, Sn, and Sb element powders, together with strict control of fabrication parameters—including accurate stoichiometry, thorough cleaning of milling equipment, and consistent ball milling and SPS conditions (temperature, pressure, and holding time). In addition, we synthesized two additional parallel batches under identical conditions, which exhibited nearly identical thermoelectric properties (**Figure R6**), confirming the robustness and reproducibility of our synthesis approach.

Figure R6. Thermoelectric performance of three batches of $\text{Zr}_{0.75}\text{Hf}_{0.25}\text{NiSn}_{0.99}\text{Sb}_{0.01}$ samples sintered at 950 °C with a holding time of 4 minutes (same as sample D95). (a) Temperature-dependent electrical conductivity (σ); (b) Temperature-dependent Seebeck coefficient (α); (c) Temperature-dependent thermal conductivity (κ); (d) Temperature-dependent dimensionless figure of merit (zT).

Regarding the lattice thermal conductivity (κ_L), among the identified minor phases, HfO_2 and ZrHfO_2 are particularly effective in reducing κ_L due to their presence as nanoprecipitates (10–20 nm), as revealed by atom probe tomography (APT). These oxide nanoprecipitates are distributed within the grain interiors and act as efficient phonon scattering centers, complementing the intrinsic phonon scattering from the host matrix. Moreover, with a total weight percent (*wt%*) of approximately 2%, these oxides represent the primary secondary phases in the material. Their nanoscale dimensions and strong interfacial mismatch significantly enhance phonon scattering, particularly for mid- to high-frequency phonons.

These points have been clarified and discussed in the revised manuscript (**Results section, pages 8 and 15**) and Supplementary Information (**Table S2 and Figure S16**).

4. Are the synthesized samples thermally stable within the measurement temperature ranges? Please provide clarification by including the heating and cooling curves. Additionally, temperature-dependent heat capacity measurements would be valuable for assessing the phase behavior within the measured/ sintering temperature range.

Response:

Thank you for pointing this out. To evaluate thermal stability, we conducted thermogravimetric analysis (TGA) and differential scanning calorimetry (DSC) on the porous sample (D95). The TGA results (**Figure R7a**) show no measurable mass change during six consecutive heating–cooling cycles from 150 °C to 700 °C, indicating good thermal stability and the absence of oxidation, sublimation, or decomposition. In addition, the temperature-dependent electrical conductivity and Seebeck coefficient show high repeatability within different measurement cycles (**Figure R4**). Furthermore, the DSC curves (**Figure R7b**) exhibit no endothermic or exothermic peaks throughout a similar temperature range, suggesting the absence of phase transitions or structural instabilities.

Figure R7. (a) TGA results of 6 heating–cooling cycles from 150 °C to 700 °C and (b) DSC curve from 35 °C to 680 °C of sample D95.

These results confirm that the sample maintains its phase integrity and composition over the entire measurement temperature range. We have included these thermal analysis curves and added relevant discussion in the revised manuscript and Supplementary Information.

5. Comment on the measurement error in sound velocity. Similarly, discuss the propagation errors and their impact on the accuracy of the zT measurements for the synthesized alloys.

Response:

We appreciate your insightful comment regarding the measurement uncertainty. The room-temperature sound velocity was measured using the RITEC Advanced Ultrasonic Measurement System RAM-5000. The primary sources of uncertainty arise from (1) sample thickness measurements, with a typical error of $\pm 1\text{--}2\%$, and (2) time-of-flight determination, which benefits from the high precision of the system, with a reported timing error of $<0.1\%$ (Handbook of Advanced Measurement System, RAM-5000, RITEC Inc.). Taking both factors into account, the overall uncertainty in sound velocity is estimated to be within $\pm 2\%$.

Regarding the figure of merit (zT), we acknowledge that it is sensitive to the propagation of measurement uncertainties in electrical conductivity ($\pm 4\%$), Seebeck coefficient ($\pm 5\%$), and thermal conductivity ($\pm 7\%$). The uncertainty in thermal conductivity arises from a $\sim 2\%$ error in density and $\sim 4\%$ in thermal diffusivity. Based on standard error propagation analysis, the resulting uncertainties in power factor and zT are estimated to be $\pm 10\%$ and $\pm 15\%$, respectively.

It is important to emphasize, however, that the thermoelectric property measurements in this study were conducted using commercial instruments under controlled conditions. This consistency reduces systematic discrepancies and ensures that relative comparisons among the different samples remain accurate and valid. We have included the discussion of these uncertainties in the revised manuscript (**Methods section**), in accordance with your comment.

6. The authors have apparently overlooked recent studies on high-performance ZrNiSn half-Heusler alloys, where a $zT > 1$ was achieved through various doping strategies, particularly with Hf substitution and role of Ni interstitials. Such omission weakens the comparison between the synthesized samples and the current state of the art.

Response:

Thank you for this important observation, and we sincerely apologize for the earlier oversight. We have now revised the zT plot (Figure R8) to include several recent high-performance ZrNiSn-based half-Heusler studies, particularly those employing Hf substitution, optimized doping strategies, and the tuning of Ni interstitial concentrations. These updates provide a more comprehensive comparison with the current state of the art.

As shown in the updated benchmark, our sample achieves a peak zT of 1.33 at 873 K, which is comparable to or exceeds most of the ZrNiSn-based materials reported to date. What distinguishes our work is that this high performance is achieved using a simplified, scalable

reactive sintering approach. Unlike many literature reports that rely on high-temperature melting, extensive doping, or post-synthesis treatments (e.g., long-duration annealing), our method enables the formation of beneficial hierarchical microstructures through a direct, solid-state route. This approach not only simplifies the synthesis process but also improves its scalability and reproducibility.

Figure R8. Comparison of zT values of our $\text{Zr}_{0.75}\text{Hf}_{0.25}\text{NiSn}_{0.99}\text{Sb}_{0.01}$ with those of previously reported MNiSn-based materials synthesized using conventional methods (*Mater. Today*, 2020, 36, 63; *Nat. Commun.*, 2024, 15, 5978; *Mater. Today Phys.*, 2023, 33, 101049; *Adv. Energy Mater.*, 2024, 14, 2402399; *Adv. Energy Mater.*, 2013, 3, 1210; *Adv. Mater.*, 2017, 29, 1702091; *J. Am. Chem. Soc.*, 2011, 133, 18843; *Energy Environ. Sci.*, 2019, 12, 3390; *Adv. Energy Mater.*, 2020, 10, 2000888).

These revisions better position our work within the broader research landscape and emphasize its unique contributions. We have updated both Figure 1 and the accompanying discussion in the revised manuscript to reflect this more complete and accurate comparison.

7. There are prevailing typos and errors which need to be thoroughly checked and corrected. For instance, “Piont defects decrease” in Figure 3c. Unit of temperature should be in Kelvin.

Response:

Thank you for your careful review. We have thoroughly checked the manuscript and corrected all typographical, grammatical, and formatting errors. Regarding the temperature units, we

would like to clarify that the x -axis in Figure 3c denotes the sintering temperature, which is intentionally expressed in degrees Celsius ($^{\circ}\text{C}$) to distinguish it from the measurement temperature for thermoelectric properties, which is expressed in Kelvin (K).

Once again, we sincerely thank you for your thorough evaluation and constructive suggestions. Your insightful comments have significantly improved the clarity, rigor, and overall quality of our manuscript.

Reviewer #3 (Remarks to the Author): Comments to the authors (Manuscript id: NCOMMS-25-26960)

This manuscript presents a novel approach to synthesize ZrNiSn-based half-Heusler compounds with multi-scale architectures comprising interstitial defects, grain boundaries, nanoprecipitates, and pores, enabling strong phonon scattering. Thus, significantly low lattice thermal conductivity $1.9 \text{ W m}^{-1} \text{ K}^{-1}$ and a high power factor of $50 \mu\text{W cm}^{-1} \text{ K}^{-2}$ leading to a notably high thermoelectric figure of merit $(zT) = 1.33$ at 873 K is reported in $\text{Zr}_{0.75}\text{Hf}_{0.25}\text{NiSn}_{0.99}\text{Sb}_{0.01}$. A thorough investigation on microstructure and its impact on thermoelectric properties is provided by the authors. The results are well supported by experimental as well as theoretical investigations. This work may guide the future research for achieving low thermal conductivity and optimizing thermoelectric performance in half-Heusler compounds. Therefore, I recommend this manuscript to be published in Nature Communications after minor revision.

Response:

We sincerely thank you for your positive and encouraging comments on our work. We greatly appreciate your recognition of our strategy to engineer multi-scale architectures in ZrNiSn-based half-Heusler compounds and its impact on optimizing thermoelectric performance. All suggested revisions have been carefully addressed, and the manuscript has been updated accordingly.

1. Fig. S8; why are Sn and Sb distributed uniformly in oxide regions like the matrix?

Response:

Thank you for this insightful question. The elemental distributions in Fig. S8 were obtained using atom probe tomography (APT), which projects three-dimensional compositional information into two dimensions. The oxide precipitates are not continuous but are embedded

within the surrounding matrix. As a result, the apparent homogeneous distribution of Sn and Sb arises from this projection effect, rather than indicating a truly uniform spatial distribution. To verify the internal composition, we extracted a 10 nm-thick slice through the precipitates and confirmed that elements such as Sn and Sb are absent within them, as shown in **Figure R9**.

Figure R9. Atom probe tomography (APT) reconstruction and elemental distributions of Zr, Hf, Ni, Sn, Sb and oxides of Zr and Hf for $Zr_{0.75}Hf_{0.25}NiSn_{0.99}Sb_{0.01}$ sample sintered at 950 °C, with a 10 nm-thick slice through the precipitates taken from the sample.

We have added clarification of this point in the revised **Supplementary Information (Figure S8)** to verify the internal composition.

- Page #9, line, “However, at 873 K, where dipolar diffusion occurs in all samples...”. Please correct typo, dipolar should be bipolar.

Response:

We thank the reviewer for catching this typographical error. The term “dipolar” on **page 9** has been corrected to “bipolar” as suggested. We appreciate your attention to detail, and the manuscript has been revised accordingly.

- Page #10, paragraph, “To investigate the influence...”. Please check and correct figure numbers (Figure S10a-d) and (Figure S10e and f).

Response:

Thank you very much for your careful review. We have checked and corrected the figure references in the manuscript as suggested. The correct figure numbers are now **Figure S10a and Figure S10b and c**, and the text has been updated accordingly.

4. Page #13, line, “The temperature dependence ...from -0.27 to -0.16 between 10 K and 300 K (Figure S12).” It should be the sintering temperature instead of measurement temperature (10 K and 300 K), please check and correct it.

Response:

Thank you for pointing out this important oversight. You are absolutely correct—the temperature range in this context refers to the sintering temperature, not the measurement temperature. We have revised the sentence on **page 13** to correct this error. We appreciate your attention to detail, which has helped improve the accuracy and clarity of the manuscript.

5. Fig. 5b and S13b; why do porous samples (such as D95) show a sharp rise in the Seebeck coefficient at high temperature compared to high density samples? This is compensating for the electrical conductivity reduced due to porosity and thus leading to a high-power factor as well as high zT . Therefore, an explanation is needed in the manuscript.

Response:

Thank you for your thoughtful comment. The sharp increase in the Seebeck coefficient at high temperatures observed in porous samples such as D95 can be primarily attributed to an enhanced energy filtering effect induced by the presence of pores and interfacial structures. These structural features introduce energy barriers that preferentially scatter low-energy charge carriers while allowing higher-energy carriers to contribute to electrical transport. This selective scattering raises the average energy of the charge carriers, thereby increasing the Seebeck coefficient. Importantly, the theoretical framework underpinning this mechanism was available from (*J. Appl. Phys.* 2010, 107, 094308), which models electron scattering by nanoscale pores in terms of the quantum mechanical approach.

Subsequent experimental studies have further validated this effect in porous and nanostructured systems interfaces (*Mater. Today* 2020, 36, 63–72; *Adv. Funct. Mater.* 2020, 30, 1901789; *ACS Appl. Energy Mater.* 2021, 4, 1915–1923; *Nano Energy* 2019, 60, 1–7). While porosity inherently reduces electrical conductivity due to disrupted carrier pathways, the enhanced Seebeck coefficient compensates for this loss to some extent. Consequently, the power factor

remains relatively high. When combined with the reduced thermal conductivity typical of porous materials, this leads to an overall enhancement in zT .

We have added a corresponding explanation in the revised manuscript (**Results section, page 14**) to clarify this behavior.

Final Remark:

We would like to express our sincere gratitude to all three reviewers for your thoughtful comments and constructive suggestions. Your careful evaluations and insightful feedback have been instrumental in improving the clarity, depth, and overall quality of our manuscript. We greatly appreciate the time and effort you devoted to reviewing our work.

Dear Reviewers,

We sincerely thank you for your time and effort in reviewing our manuscript entitled “*High-Performance ZrNiSn-based Half-Heusler Thermoelectrics with Hierarchical Architectures Enabled by Reactive Sintering*” (NCOMMS-25-26960A). We greatly appreciate your positive evaluation of our work and valuable feedback. In response, we have carefully revised the manuscript. For clarity, your feedback are presented in **blue**, followed by our point-by-point responses in **black**. The revisions made to the manuscript are highlighted in **red** in the revised files.

Reviewer #1 (Remarks to the Author):

All issues have been adequately addressed. I recommend acceptance of the manuscript.

Response:

We sincerely appreciate your thorough and positive evaluation of our work.

Reviewer #2 (Remarks to the Author):

The revised manuscript satisfactorily addresses many of the previous comments. However, Figure 5f, which compares the average thermoelectric figure of merit (zT_{avg}) of $\text{Zr}_{0.75}\text{Hf}_{0.25}\text{NiSn}_{0.99}\text{Sb}_{0.01}$ with other MNiSn-based compounds, overlooks several recent studies that report zT_{avg} values exceeding 1 using conventional synthesis routes. Such omissions limit the impact and relevance of comparative analysis, and it is encouraged to incorporate more recent and higher-performing data points to provide a more comprehensive perspective.

Response:

We sincerely appreciate your constructive feedback. We have incorporated recent data points to **Figure 5f**, as shown below.

Figure 5f. Comparison of average zT (zT_{average}) values of our $\text{Zr}_{0.75}\text{Hf}_{0.25}\text{NiSn}_{0.99}\text{Sb}_{0.01}$ with those of previously reported MNiSn-based materials synthesized using conventional methods (*Nat. Commun.*, 2024, 15, 5978; *Adv. Energy Mater.*, 2024, 14, 2402399; *Mater. Today*, 2020, 36, 63; *Adv. Energy Mater.*, 2013, 3, 1210; *Adv. Mater.*, 2017, 29, 1702091; *Energy Environ. Sci.*, 2019, 12, 3390; *Adv. Energy Mater.*, 2020, 10, 2000888; *Mater. Today Phys.*, 2023, 33, 101049).

Thank you again for this valuable suggestion, which improves the quality of our manuscript.

Reviewer #3 (Remarks to the Author):

The authors have responded to all of my previous comments in a satisfactory manner. The revised manuscript is clear, well-structured, and scientifically robust. I find no remaining issues and recommend the manuscript for acceptance in its current form.

Response:

We sincerely appreciate your thoughtful feedback and recognition of the manuscript.